# 3D-Aware Scene Manipulation via Inverse Graphics

**Shunyu Yao***
IIIS, Tsinghua University

**Tzu-Ming Harry Hsu***
MIT CSAIL

**Jun-Yan Zhu**
MIT CSAIL

**Jiajun Wu**
MIT CSAIL

**Antonio Torralba**
MIT CSAIL

**William T. Freeman**
MIT CSAIL, Google Research

**Joshua B. Tenenbaum**
MIT CSAIL

## Abstract

We aim to obtain an interpretable, expressive, and disentangled scene representation that contains comprehensive structural and textural information for each object. Previous scene representations learned by neural networks are often uninterpretable, limited to a single object, or lacking 3D knowledge. In this work, we propose 3D scene de-rendering networks (3D-SDN) to address the above issues by integrating disentangled representations for semantics, geometry, and appearance into a deep generative model. Our scene encoder performs inverse graphics, translating a scene into a structured object-wise representation. Our decoder has two components: a differentiable shape renderer and a neural texture generator. The disentanglement of semantics, geometry, and appearance supports 3D-aware scene manipulation, e.g., rotating and moving objects freely while keeping the consistent shape and texture, and changing the object appearance without affecting its shape. Experiments demonstrate that our editing scheme based on 3D-SDN is superior to its 2D counterpart.

## 1   Introduction

Humans are incredible at perceiving the world, but more distinguishing is our mental ability to simulate and imagine what will happen. Given a street scene as in Fig. 1, we can effortlessly detect and recognize cars and their attributes, and more interestingly, imagine how cars may move and rotate in the 3D world. Motivated by such human abilities, in this work we seek to obtain an interpretable, expressive, and disentangled scene representation for machines, and employ the learned representation for flexible, 3D-aware scene manipulation.

Deep generative models have led to remarkable breakthroughs in learning hierarchical representations of images and decoding them back to images [Goodfellow et al., 2014]. However, the obtained representation is often limited to a single object, hard to interpret, and missing the complex 3D structure behind visual input. As a result, these deep generative models cannot support manipulation tasks such as moving an object around as in Fig. 1. On the other hand, computer graphics engines use a predefined, structured, and disentangled input (e.g., graphics code), often intuitive for scene manipulation. However, it is in general intractable to infer the graphics code from an input image.

In this paper, we propose 3D scene de-rendering networks (3D-SDN) to incorporate an object-based, interpretable scene representation into a deep generative model. Our method employs an encoder-decoder architecture and has three branches: one for scene semantics, one for object geometry and 3D pose, and one for the appearance of objects and the background. As shown in Fig. 2, the semantic de-renderer aims to learn the semantic segmentation of a scene. The geometric de-renderer learns to infer the object shape and 3D pose with the help of a differentiable shape renderer. The textural

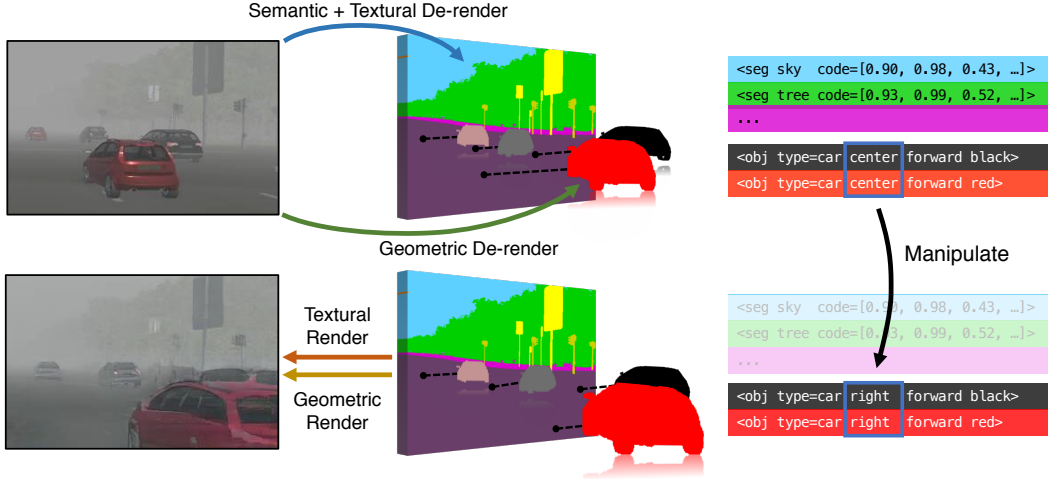

Figure 1: We propose to learn a holistic scene representation that encodes scene semantics as well as 3D and textural information. Our encoder-decoder framework learns disentangled representations for image reconstruction and 3D-aware image editing. For example, we can move cars to various locations with new 3D poses.

de-renderer learns to encode the appearance of each object and background segment. We then employ the geometric renderer and the textural renderer to recover the input scene using the above semantic, geometric, and textural information. Disentangling 3D geometry and pose from texture enables 3D-aware scene manipulation. For example in Fig. 1, to move a car closer, we can edit its position and 3D pose, but leave its texture representation untouched.

Both quantitative and qualitative results demonstrate the effectiveness of our method on two datasets: Virtual KITTI [Gaidon et al., 2016] and Cityscapes [Cordts et al., 2016]. Furthermore, we create an image editing benchmark on Virtual KITTI to evaluate our editing scheme against 2D baselines. Finally, we investigate our model design by evaluating the accuracy of obtained internal representation. Please check out our code and website for more details.

## 2    Related Work

**Interpretable image representation.** Our work is inspired by prior work on obtaining interpretable visual representations with neural networks [Kulkarni et al., 2015, Chen et al., 2016]. To achieve this goal, DC-IGN [Kulkarni et al., 2015] freezes a subset of latent codes while feeding images that move along a specific direction on the image manifold. A recurrent model [Yang et al., 2015] learns to alter disentangled latent factors for view synthesis. InfoGAN [Chen et al., 2016] proposes to disentangle an image into independent factors without supervised data. Another line of approaches are built on intrinsic image decomposition [Barrow and Tenenbaum, 1978] and have shown promising results on faces [Shu et al., 2017] and objects [Janner et al., 2017].

While prior work focuses on a single object, we aim to obtain a holistic scene understanding. Our work most resembles the paper by Wu et al. [2017a], who propose to 'de-render' an image with an encoder-decoder framework that uses a neural network as the encoder and a graphics engine as the decoder. However, their method cannot back-propagate the gradients from the graphics engine or generalize to a new environment, and the results were limited to simple game environments such as Minecraft. Unlike Wu et al. [2017a], both of our encoder and decoder are differentiable, making it possible to handle more complex natural images.

**Deep generative models.** Deep generative models [Goodfellow et al., 2014] have been used to synthesize realistic images and learn rich internal representations. Representations learned by these methods are typically hard for humans to interpret and understand, often ignoring the 3D nature of our visual world. Many recent papers have explored the problem of 3D reconstruction from a single color image, depth map, or silhouette [Choy et al., 2016, Kar et al., 2015, Tatarchenko et al., 2016,

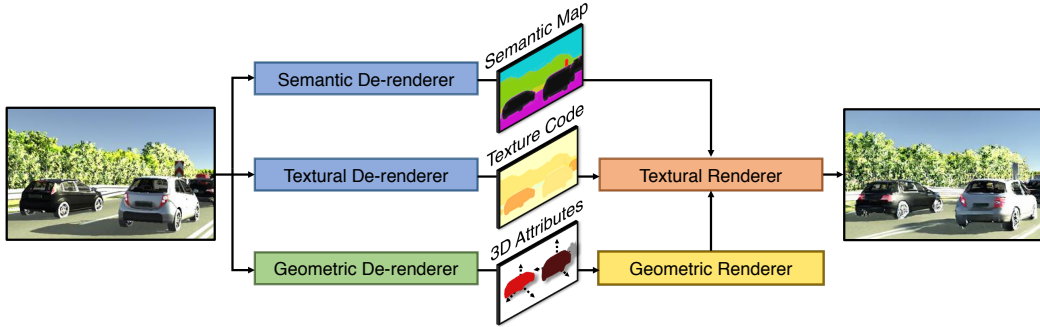

Figure 2: **Framework overview.** The de-renderer (encoder) consists of a semantic-, a textural- and a geometric branch. The textural renderer and geometric renderer then learn to reconstruct the original image from the representations obtained by the encoder modules.

Tulsiani et al., 2017, Wu et al., 2017b, 2016b, Yan et al., 2016b, Soltani et al., 2017]. Our model builds upon and extends these approaches. We infer the 3D object geometry with neural nets and re-render the shapes into 2D with a differentiable renderer. This improves the quality of the generated results and allows 3D-aware scene manipulation.

**Deep image manipulation.** Learning-based methods have enabled various image editing tasks, such as style transfer [Gatys et al., 2016], image-to-image translation [Isola et al., 2017, Zhu et al., 2017a, Liu et al., 2017], automatic colorization [Zhang et al., 2016], inpainting [Pathak et al., 2016], attribute editing [Yan et al., 2016a], interactive editing [Zhu et al., 2016], and denoising [Gharbi et al., 2016]. Different from prior work that operates in a 2D setting, our model allows 3D-aware image manipulation. Besides, while the above methods often require a given structured representation (e.g., label map [Wang et al., 2018]) as input, our algorithm can learn an internal representation suitable for image editing by itself. Our work is also inspired by previous semi-automatic 3D editing systems [Karsch et al., 2011, Chen et al., 2013, Kholgade et al., 2014]. While these systems require human annotations of object geometry and scene layout, our method is fully automatic.

## 3 Method

We propose **3D scene de-rendering networks** (3D-SDN) in an encoder-decoder framework. As shown in Fig. 2, we first de-render (encode) an image into disentangled representations for semantic, textural, and geometric information. Then, a renderer (decoder) reconstructs the image from the representation.

The semantic de-renderer learns to produce the semantic segmentation (e.g. trees, sky, road) of the input image. The 3D geometric de-renderer detects and segments objects (cars and vans) from image, and infers the geometry and 3D pose for each object with a differentiable shape renderer. After inference, the geometric renderer computes an instance map, a pose map, and normal maps for objects in the scene for the textural branch. The textural de-renderer first fuses the semantic map generated by the semantic branch and the instance map generated by the geometric branch into an instance-level semantic label map, and learns to encode the color and texture of each instance (object or background semantic class) into a texture code. Finally, the textural renderer combines the instance-wise label map (from the textural de-renderer), textural codes (from the textural de-renderer), and 3D information (instance, normal, and pose maps from the geometric branch) to reconstruct the input image.

### 3.1 3D Geometric Inference

Fig. 3 shows the 3D geometric inference module for the 3D-SDN. We first segment object instances with Mask-RCNN [He et al., 2017]. For each object, we infer its 3D mesh model and other attributes from its masked image patch and bounding box.

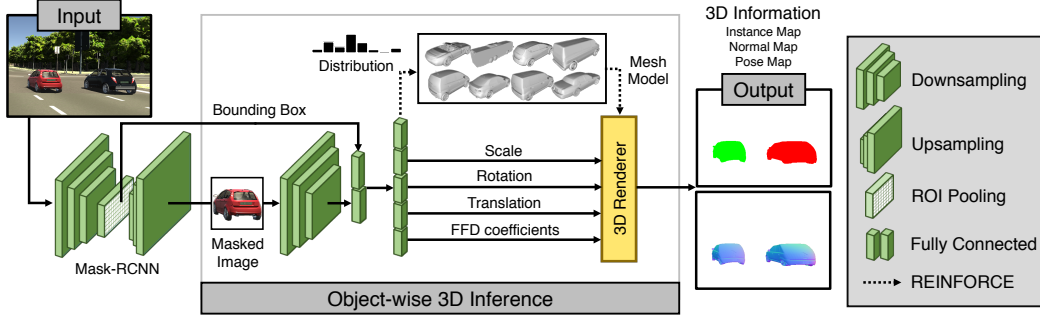

Figure 3: **3D geometric inference.** Given a masked object image and its bounding box, the geometric branch of the 3D-SDN predicts the object's mesh model, scale, rotation, translation, and the free-form deformation (FFD) coefficients. We then compute 3D information (instance map, normal maps, and pose map) using a differentiable renderer [Kato et al., 2018].

**3D estimation.** We describe a 3D object with a mesh $M$, its scale $\mathbf{s} \in \mathbb{R}^3$, rotation $\mathbf{q} \in \mathbb{R}^4$ as an unit quaternion, and translation $\mathbf{t} \in \mathbb{R}^3$. For most real-world scenarios such as road scenes, objects often lie on the ground. Therefore, the quaternion has only one rotational degree of freedom: i.e., $\mathbf{q} \in \mathbb{R}$.

As shown in Fig. 3, given an object's masked image and estimated bounding box, the geometric de-renderer learns to predict the mesh $M$ by first selecting a mesh from eight candidate shapes, and then applying a Free-Form Deformation (FFD) [Sederberg and Parry, 1986] with inferred grid point coordinates $\phi$. It also predicts the scale, rotation, and translation of the 3D object. Below we describe the training objective for the network.

**3D attribute prediction loss.** The geometric de-renderer directly predicts the values of scale $\mathbf{s}$ and rotation $\mathbf{q}$. For translation $\mathbf{t}$, it instead predicts the object's distance to the camera $t$ and the image-plane 2D coordinates of the object's 3D center, denoted as $[x_{3D}, y_{3D}]$. Given the intrinsic camera matrix, we can calculate $\mathbf{t}$ from $t$ and $[x_{3D}, y_{3D}]$. We parametrize $t$ in the log-space [Eigen et al., 2014]. As determining $t$ from the image patch of the object is under-constrained, our model predicts a normalized distance $\tau = t\sqrt{wh}$, where $[w, h]$ is the width and height of the bounding box. This reparameterization improves results as shown in later experiments (Sec. 4.2). For $[x_{3D}, y_{3D}]$, we follow the prior work [Ren et al., 2015] and predict the offset $\mathbf{e} = [(x_{3D} - x_{2D})/w, (y_{3D} - y_{2D})/h]$ relative to the estimated bounding box center $[x_{2D}, y_{2D}]$. The 3D attribute prediction loss for scale, rotation, and translation can be calculated as

$$\mathcal{L}_{\text{pred}} = \|\log \tilde{\mathbf{s}} - \log \mathbf{s}\|_2^2 + \left(1 - (\tilde{\mathbf{q}} \cdot \mathbf{q})^2\right) + \|\tilde{\mathbf{e}} - \mathbf{e}\|_2^2 + (\log \tilde{\tau} - \log \tau)^2, \qquad (1)$$

where $\tilde{\cdot}$ denotes the predicted attributes.

**Reprojection consistency loss.** We also use a reprojection loss to ensure the 2D rendering of the predicted shape fits its silhouette $\mathbf{S}$ [Yan et al., 2016b, Rezende et al., 2016, Wu et al., 2016a, 2017b]. Fig. 4a and Fig. 4b show an example. Note that for mesh selection and deformation, the reprojection loss is the only training signal, as we do not have a ground truth mesh model.

We use a differentiable renderer [Kato et al., 2018] to render the 2D silhouette of a 3D mesh $M$, according to the FFD coefficients $\phi$ and the object's scale, rotation and translation $\tilde{\pi} = \{\tilde{\mathbf{s}}, \tilde{\mathbf{q}}, \tilde{\mathbf{t}}\}$: $\tilde{\mathbf{S}} = \text{RenderSilhouette}(\text{FFD}_\phi(M), \tilde{\pi})$. We then calculate the reprojection loss as $\mathcal{L}_{\text{reproj}} = \left\|\tilde{\mathbf{S}} - \mathbf{S}\right\|$. We ignore the region occluded by other objects. The full loss function for the geometric branch is thus $\mathcal{L}_{\text{pred}} + \lambda_{\text{reproj}}\mathcal{L}_{\text{reproj}}$, where $\lambda$ controls the relative importance of two terms.

**3D model selection via REINFORCE.** We choose the mesh $M$ from a set of eight meshes to minimize the reprojection loss. As the model selection process is non-differentiable, we formulate the model selection as a reinforcement learning problem and adopt a multi-sample REINFORCE paradigm [Williams, 1992] to address the issue. The network predicts a multinomial distribution over the mesh models. We use the negative reprojection loss as the reward. We experimented with a single mesh without FFD in Fig. 4c. Fig. 4d shows a significant improvement when the geometric branch learns to select from multiple candidate meshes and allows flexible deformation.

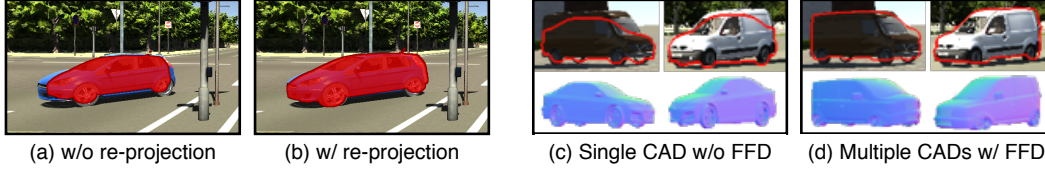

| (a) w/o re-projection | (b) w/ re-projection | (c) Single CAD w/o FFD | (d) Multiple CADs w/ FFD |

Figure 4: (a)(b) Re-projection consistency loss: Object silhouettes rendered without and with re-projection consistency loss. (c)(d) Multiple CAD models and free form deformation (FFD): In (c), a generic car model without FFD fails to represent the input vans. In (d), our model learns to choose the best-fitting mesh from eight candidate meshes and allows FFD. As a result, we can reconstruct the silhouettes more precisely.

## 3.2    Semantic and Textural Inference

The semantic branch of the 3D-SDN uses a semantic segmentation model DRN [Yu et al., 2017, Zhou et al., 2017] to obtain an semantic map of the input image. The textural branch of the 3D-SDN first obtains an instance-wise semantic label map $\mathbf{L}$ by combining the semantic map generated by the semantic branch and the instance map generated by the geometric branch, resolving any conflict in favor of the instance map [Kirillov et al., 2018]. Built on recent work on multimodal image-to-image translation [Zhu et al., 2017b, Wang et al., 2018], our textural branch encodes the texture of each instance into a low dimensional latent code, so that the textural renderer can later reconstruct the appearance of the original instance from the code. By 'instance' we mean a background semantic class (e.g., road, sky) or a foreground object (e.g., car, van). Later, we combine the object textural code with the estimated 3D information to better reconstruct objects.

Formally speaking, given an image $\mathbf{I}$ and its instance label map $\mathbf{L}$, we want to obtain a feature embedding $\mathbf{z}$ such that $(\mathbf{L}, \mathbf{z})$ can later reconstruct $\mathbf{I}$. We formulate the textural branch of the 3D-SDN under a conditional adversarial learning framework with three networks $(G, D, E)$: a textural de-renderer $E : (\mathbf{L}, \mathbf{I}) \to \mathbf{z}$, a texture renderer $G : (\mathbf{L}, \mathbf{z}) \to \mathbf{I}$ and a discriminator $D : (\mathbf{L}, \mathbf{I}) \to [0, 1]$ are trained jointly with the following objectives.

To increase the photorealism of generated images, we use a standard conditional GAN loss [Goodfellow et al., 2014, Mirza and Osindero, 2014, Isola et al., 2017] as: *

$$\mathcal{L}_{\text{GAN}}(G, D, E) = \mathbb{E}_{\mathbf{L}, \mathbf{I}} \Big[ \log \left( D(\mathbf{L}, \mathbf{I}) \right) + \log \left( 1 - D \left( \mathbf{L}, \tilde{\mathbf{I}} \right) \right) \Big], \tag{2}$$

where $\tilde{\mathbf{I}} = G(\mathbf{L}, E(\mathbf{L}, \mathbf{I}))$ is the reconstructed image. To stabilize the training, we follow the prior work [Wang et al., 2018] and use both discriminator feature matching loss [Wang et al., 2018, Larsen et al., 2016] and perceptual loss [Dosovitskiy and Brox, 2016, Johnson et al., 2016], both of which aim to match the statistics of intermediate features between generated and real images:

$$\mathcal{L}_{\text{FM}}(G, D, E) = \mathbb{E}_{\mathbf{L}, \mathbf{I}} \left[ \sum_{i=1}^{T_F} \frac{1}{N_i} \left\| F^{(i)}(\mathbf{I}) - F^{(i)} \left( \tilde{\mathbf{I}} \right) \right\|_1 + \sum_{i=1}^{T_D} \frac{1}{M_i} \left\| D^{(i)}(\mathbf{I}) - D^{(i)} \left( \tilde{\mathbf{I}} \right) \right\|_1 \right], \tag{3}$$

where $F^{(i)}$ denotes the $i$-th layer of a pre-trained VGG network [Simonyan and Zisserman, 2015] with $N_i$ elements. Similarly, for our our discriminator $D$, $D^{(i)}$ denotes the $i$-th layer with $M_i$ elements. $T_F$ and $T_D$ denote the number of layers in network $F$ and $D$. We fix the network $F$ during our training. Finally, we use a pixel-wise image reconstruction loss as:

$$\mathcal{L}_{\text{Recon}}(G, E) = \mathbb{E}_{\mathbf{L}, \mathbf{I}} \left[ \left\| \mathbf{I} - \tilde{\mathbf{I}} \right\|_1 \right]. \tag{4}$$

The final training objective is formulated as a minimax game between $(G, E)$ and $D$:

$$G^*, E^* = \underset{G,E}{\arg\min} \left( \underset{D}{\max} \left( \mathcal{L}_{\text{GAN}}(G, D, E) \right) + \lambda_{\text{FM}} \mathcal{L}_{\text{FM}}(G, D, E) + \lambda_{\text{Recon}} \mathcal{L}_{\text{Recon}}(G, E) \right), \tag{5}$$

where $\lambda_{\text{FM}}$ and $\lambda_{\text{Recon}}$ control the relative importance of each term.

**Decoupling geometry and texture.** We observe that the textural de-renderer often learns not only texture but also object poses. To further decouple these two factors, we concatenate the inferred 3D information (i.e., pose map and normal map) from the geometric branch to the texture code map $\mathbf{z}$ and feed both of them to the textural renderer $G$. Also, we reduce the dimension of the texture code so that the code can focus on texture as the 3D geometry and pose are already provided. These two modifications help encode textural features that are independent of the object geometry. It also resolves ambiguity in object poses: e.g., cars share similar silhouettes when facing forward or backward. Therefore, our renderer can synthesize an object under different 3D poses. (See Fig. 5b and Fig. 7b for example).

### 3.3 Implementation Details

**Semantic branch.** Our semantic branch adopts Dilated Residual Networks (DRN) for semantic segmentation [Yu et al., 2017, Zhou et al., 2017]. We train the network for 25 epochs.

**Geometric branch.** We use Mask-RCNN for object proposal generation [He et al., 2017]. For object meshes, we choose eight CAD models from ShapeNet [Chang et al., 2015] including cars, vans, and buses. Given an object proposal, we predict its scale, rotation, translation, $4^3$ FFD grid point coefficients, and an 8-dimensional distribution across candidate meshes with a ResNet-18 network [He et al., 2015]. The translation $\mathbf{t}$ can be recovered using the estimated offset $\mathbf{e}$, the normalized distance $\log \tau$, and the ground truth focal length of the image. They are then fed to a differentiable renderer [Kato et al., 2018] to render the instance map and normal map.

We empirically set $\lambda_{\mathrm{reproj}} = 0.1$. We first train the network with $\mathcal{L}_{\mathrm{pred}}$ using Adam [Kingma and Ba, 2015] with a learning rate of $10^{-3}$ for 256 epochs and then fine-tune the model with $\mathcal{L}_{\mathrm{pred}} + \lambda_{\mathrm{reproj}} \mathcal{L}_{\mathrm{reproj}}$ and REINFORCE with a learning rate of $10^{-4}$ for another 64 epochs.

**Textural branch.** We first train the semantic branch and the geometric branch separately and then train the textural branch using the input from the above two branches. We use the same architecture as in Wang et al. [2018]. We use two discriminators of different scales and one generator. We use the VGG network [Simonyan and Zisserman, 2015] as the feature extractor $F$ for loss $\lambda_{\mathrm{FM}}$ (Eqn. 3). We set the dimension of the texture code as 5. We quantize the object's rotation into 24 bins with one-hot encoding and fill each rendered silhouette of the object with its rotation encoding, yielding a pose map of the input image. Then we concatenate the pose map, the predicted object normal map, the texture code map $\mathbf{z}$, the semantic label map, and the instance boundary map together, and feed them to the neural textural renderer to reconstruct the input image. We set $\lambda_{\mathrm{FM}} = 5$ and $\lambda_{\mathrm{Recon}} = 10$, and train the textural branch for 60 epochs on Virtual KITTI and 100 epochs on Cityscapes.

## 4 Results

We report our results in two parts. First, we present how the 3D-SDN enables 3D-aware image editing. For quantitative comparison, we compile a Virtual KITTI image editing benchmark to contrast 3D-SDNs and baselines without 3D knowledge. Second, we analyze our design choices and evaluate the accuracy of representations obtained by different variants. The code and full results can be found at our website.

**Datasets.** We conduct experiments on two street scene datasets: Virtual KITTI [Gaidon et al., 2016] and Cityscapes [Cordts et al., 2016]. Virtual KITTI serves as a proxy to the KITTI dataset [Geiger et al., 2012]. The dataset contains five virtual worlds, each rendered under ten different conditions, leading to a sum of 21,260 images. For each world, we use either the first or the last $80\%$ consecutive frames for training and the rest for testing. For object-wise evaluations, we use objects with more than 256 visible pixels, a $< 70\%$ occlusion ratio, and a $< 70\%$ truncation ratio, following the ratios defined in Gaidon et al. [2016]. In our experiments, we downscale Virtual KITTI images to $624 \times 192$ and Cityscapes images to $512 \times 256$.

We have also built the *Virtual KITTI Image Editing Benchmark*, allowing us to evaluate image editing algorithms systematically. The benchmark contains 92 pairs of images in the test set with the camera either stationary or almost still. Fig. 1 shows an example pair. For each pair, we formulate the edit with object-wise operations. Each operation is parametrized by a starting position $(x_{3D}^{\mathrm{src}}, y_{3D}^{\mathrm{src}})$, an

Original image                    Edited images

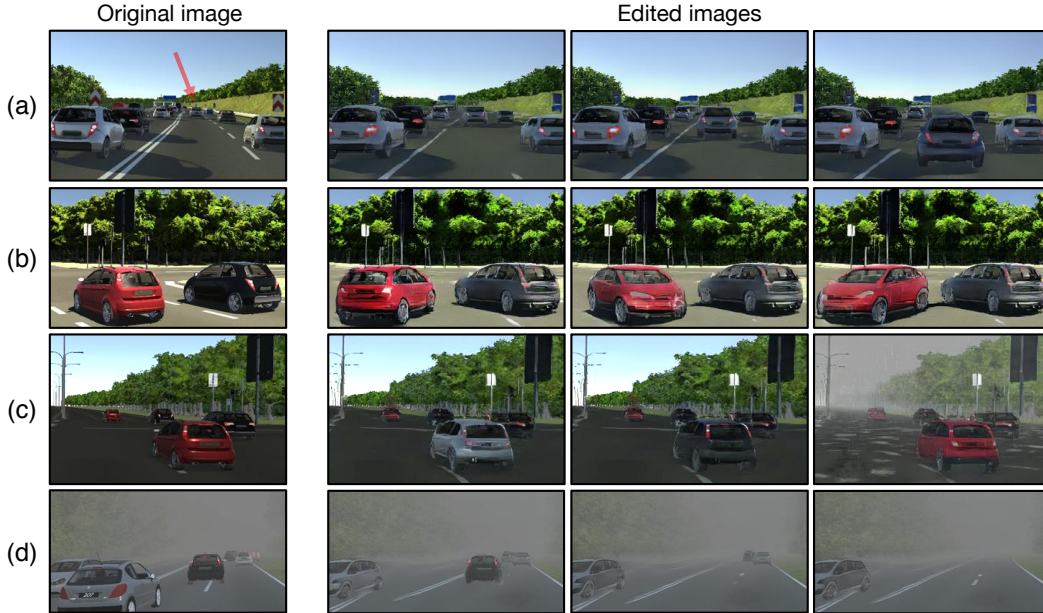

Figure 5: **Example user editing results on Virtual KITTI.** (a) We move a car closer to the camera, keeping the same texture. (b) We can synthesize the same car with different 3D poses. The same texture code is used for different poses. (c) We modify the appearance of the input red car using new texture codes. Note that its geometry and pose stay the same. We can also change the environment by editing the background texture codes. (d) We can inpaint occluded regions and remove objects.

|                  | 3D-SDN (ours) | 2D     | 2D+    |
| ---------------- | ------------- | ------ | ------ |
| LPIPS (whole)    | **0.1280**    | 0.1316 | 0.1317 |
| LPIPS (all)      | **0.1444**    | 0.1782 | 0.1799 |
| LPIPS (largest)  | **0.1461**    | 0.1795 | 0.1813 |

(a) Perception similarity scores

|               | 2D     | 2D+    |
| ------------- | ------ | ------ |
| 3D-SDN (ours) | 76.88% | 74.28% |

(b) Human study results

Table 1: **Evaluations on Virtual KITTI editing benchmark.** (a) We evaluate the perceptual similarity [Zhang et al., 2018] on the whole image (whole), all edited regions (all) of the image, and the largest edited region (largest) of the image, respectively. Lower scores are better. (b) Human subjects compare our method against two baselines. The percentage shows how often they prefer 3D-SDNs to the baselines. Our method outperforms previous 2D approaches consistently.

ending position $\left(x_{3D}^{\text{tgt}}, y_{3D}^{\text{tgt}}\right)$ (both are object's 3D center in image plane), a zoom-in factor $\rho$, and a rotation $\Delta r_y$ with respect to the $y$-axis of the camera coordinate system.

The Cityscapes dataset contains 2,975 training images with pixel-level semantic segmentation and instance segmentation ground truth, but with no 3D annotations, making the geometric inference more challenging. Therefore, given each image, we first predict 3D attributes with our geometric branch pre-trained on Virtual KITTI dataset; we then optimize both attributes and mesh parameters $\pi$ and $\phi$ by minimizing the reprojection loss $\mathcal{L}_{\text{reproj}}$. We use the Adam solver [Kingma and Ba, 2015] with a learning rate of 0.03 for 16 iterations.

## 4.1   3D-Aware Image Editing

The semantic, geometric, and textural disentanglement provides an expressive 3D image manipulation scheme. We can modify the 3D attributes of an object to translate, scale, or rotate it in the 3D world, while keeping the consistent visual appearance. We can also change the appearance of the object or the background by modifying the texture code alone.

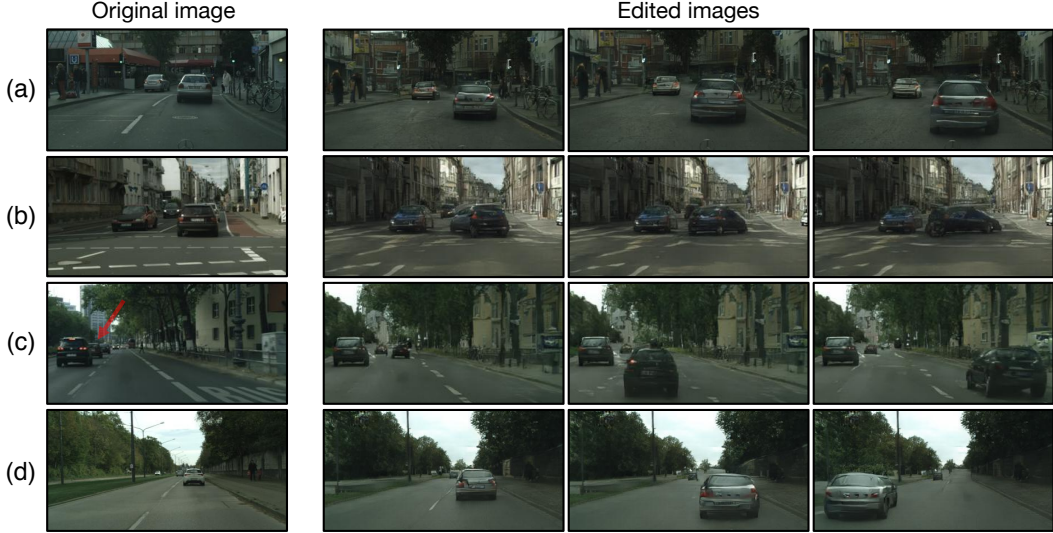

Figure 6: **Example user editing results on Cityscapes.** (a) We move two cars closer to the camera. (b) We rotate the car with different angles. (c) We recover a tiny and occluded car and move it closer. Our model can synthesize the occluded region as well as view the occluded car from the side. (d) We move a small car closer and then change its locations.

**Methods.** We compare our 3D-SDNs with the following two baselines:

- 2D: Given the source and target positions, the naïve 2D baseline only applies the 2D translation and scaling, discarding the $\Delta r_y$ rotation.

- 2D+: The 2D+ baseline includes the 2D operations above and rotates the 2D silhouette (instead of the 3D shape) along the $y$-axis according to the rotation $\Delta r_y$ in the benchmark.

**Metrics.** The pixel-level distance might not be a meaningful similarity metric, as two visually similar images may have a large L1/L2 distance [Isola et al., 2017]. Instead, we adopt the Learned Perceptual Image Patch Similarity (LPIPS) metric [Zhang et al., 2018], which is designed to match the human perception. LPIPS ranges from 0 to 1, with 0 being the most similar. We apply LPIPS on (1) the full image, (2) all edited objects, and (3) the largest edited object.

Besides, we conduct a human study, where we show the target image as well as the edited results from two different methods: 3D-SDN vs. 2D and 3D-SDN vs. 2D+. We ask 120 human subjects on Amazon Mechnical Turk which edited result looks closer to the target. For better visualization, we highlight the largest edited object in red. We then compute, between a pair of methods, how often one method is preferred, across all test images.

**Results.** Fig. 5 and Fig. 6 show qualitative results on Virtual KITTI and Cityscapes, respectively. By modifying semantic, geometric, and texture codes, our editing interface enables a wide range of scene manipulation applications. Fig. 7 shows a direct comparison to a state-of-the-art 2D manipulation method pix2pixHD [Wang et al., 2018]. Quantitatively, Table 1a shows that our 3D-SDN outperforms both baselines by a large margin regarding LPIPS. Table 1b shows that a majority of the human subjects perfer our results to 2D baselines.

### 4.2 Evaluation on the Geometric Representation

**Methods.** As described in Section 3.1, we adopt multiple strategies to improve the estimation of 3D attributes. As an ablation study, we compare the full 3D-SDN, which is first trained using $\mathcal{L}_{pred}$ then fine-tuned using $\mathcal{L}_{pred} + \lambda_{reproj}\mathcal{L}_{reproj}$, with its four variants:

- w/o $\mathcal{L}_{reproj}$: we only use the 3D attribute prediction loss $\mathcal{L}_{pred}$.

- w/o quaternion constraint: we use the full rotation space characterized by a unit quaternion $\mathbf{q} \in \mathbb{R}^4$, instead of limiting to $\mathbb{R}$.

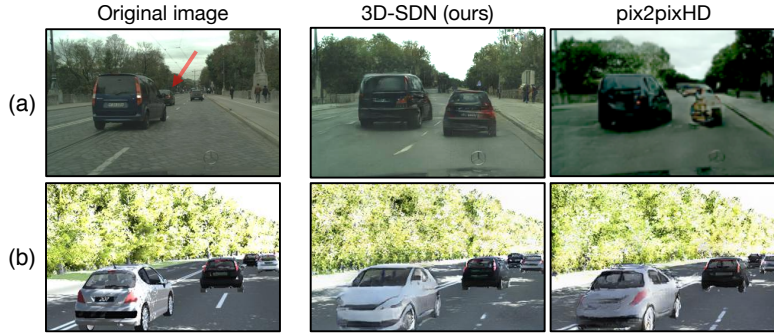

| | Original image | 3D-SDN (ours) | pix2pixHD |
|---|---|---|---|

Figure 7: **Comparison between 3D-SDN (ours) and pix2pixHD** [Wang et al., 2018]. (a) We successfully recover the mask of an occluded car and move it closer to the camera while pix2pixHD fails. (b) We rotate the car from back to front. With the texture code encoded from the back view and a frontal pose, our model can remove the tail lights, while pix2pixHD cannot, given the same instance map.

| | Orientation similarity | Distance ($\times 10^{-2}$) | Scale | Reprojection error ($\times 10^{-3}$) |
|---|---|---|---|---|
| Mousavian et al. [2017] | 0.976 | 4.41 | 0.391 | 9.80 |
| w/o $\mathcal{L}_{\text{reproj}}$ | 0.980 | 3.76 | **0.372** | 9.54 |
| w/o quaternion constraint | 0.970 | 4.59 | 0.403 | 7.58 |
| w/o normalized distance $\tau$ | 0.979 | 4.27 | 0.420 | 6.42 |
| w/o MultiCAD and FFD | 0.984 | **3.37** | 0.464 | 4.60 |
| 3D-SDN (ours) | **0.987** | 3.87 | 0.382 | **3.37** |

Table 2: **Performance of 3D attributes prediction on Virtual KITTI**. We compare our full model with its four variants. Our full model performs the best regarding most metrics. Our model obtains much lower reprojection error. Refer to the text for details about our metrics.

- w/o normalized distance $\tau$: we predict the original distance $t$ in log space rather than the normalized distance $\tau$.
- w/o MultiCAD and FFD: we use a single CAD model without free-form deformation (FFD).

We also compare with a 3D bounding box estimation method [Mousavian et al., 2017], which first infers the object's 2D bounding box and pose from input and then searches for its 3D bounding box.

**Metrics.** We use different metrics for different quantities. For rotation, we compute the orientation similarity $(1 + \cos \theta)/2$ [Geiger et al., 2012], where $\theta$ is the geodesic distance between the predicted and the ground truth rotations; for distance, we adopt an absolute logarithm error $\left| \log t - \log \tilde{t} \right|$; and for scale, we adopt the Euclidean distance $\|\mathbf{s} - \tilde{\mathbf{s}}\|_2$. In addition, we compute the per-pixel reprojection error between projected 2D silhouettes and ground truth segmentation masks.

**Results.** Table 2 shows that our full model has significantly smaller 2D reprojection error than other variants. All of the proposed components contribute to the performance.

## 5 Conclusion

In this work, we have developed 3D scene de-rendering networks (3D-SDN) to obtain an interpretable and disentangled scene representation with rich semantic, 3D structural, and textural information. Though our work mainly focuses on 3D-aware scene manipulation, the learned representations could be potentially useful for various tasks such as image reasoning, captioning, and analogy-making. Future directions include better handling uncommon object appearance and pose, especially those not in the training set, and dealing with deformable shapes such as human bodies.

**Acknowledgements.** This work is supported by NSF #1231216, NSF #1447476, NSF #1524817, ONR MURI N00014-16-1-2007, Toyota Research Institute, and Facebook.

## Footnotes

* indicates equal contributions. The work was done when Shunyu Yao was a visiting student at MIT CSAIL.

*We denote $\mathbb{E}_{\mathbf{L}, \mathbf{I}} \triangleq \mathbb{E}_{(\mathbf{L}, \mathbf{I}) \sim p_{\text{data}(\mathbf{L}, \mathbf{I})}}$ for simplicity.

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
