[Reviews · NeurIPS 2018]

Reviewer 1



The paper presents a method for editing images using deep inverse rendering (encoder/decoder): an image is decomposed into a semantic, textural and geometric representation, which then is combined to produce the original image. The intermediate representations can be edited, altering the final image based on the type of modification (eg, 3D rotation-translation-scaling, texture). I like the idea of inverse rendering for scene understanding and modification, and the paper proposes a good way of achieving it with deep architectures. Moreover, allowing the modifications of the geometric/texture representations is useful and can lead to further investigation of such edits. My main concern is about the quality of the edits. The paper claims keeping the texture consistent (L200) but this is not the case for example in Figure 5 (b). The color is similar, but the overall texture is very different. Also, regarding the pose not being polluted, do you refer to a case for example of having a car at 30 degrees and the texture of 90 degrees? It would be good to include explanatory figures (eg for L154-157). Having only 1 cad model is limited. What if the shape doesn't fit the object in the image (eg a sedan). A scaling factor can't compensate for the difference in shape. I didn't completely understand the translation parameterization (L111-L122) and why it is important. If the goal is a 3D edit of an object, why do you need to encode-decode all the elements in the scene and not only the effected (eg car+road)? Why you did not try on KITTI, so a comparison with [Mousavian] is more fair (they were trained on KITTI only?). At Section 4.2, it would be good to include an example of the baselines (at least in the supplementary). Summary. Quality: the presented results include only few examples and the quality is not good, especially compared to [Wang]. The texture and pose edits look plausible in some cases but in my opinion the claimed decomposition has not been achieved. Clarity: the paper is difficult to follow, with dense content (eg L111-L114) Originality and significance: the work is original and the problem is interesting.

Reviewer 2



This paper proposes to learn to extract scene representations by incorporating 3D models into a deep generative model. The learned inverse graphics model can be applied to 3D scene manipulation. The paper is well motivated, noting that contemporary deep generative models do not support simulation tasks involving object manipulation, which could be very important to agents interacting with and planning in real world environments. At a high level, the model works by disentangling 3D structure from texture in the encoder and the decoder models. Re-rendering can be performed by e.g. editing 3D attributes while fixing other, disentangled attributes such as texture. - Can the model handle occluded objects? - L117 - what does it mean that “the distance conditioned on the patch is not a well distributed quantity”? Do you mean the patch resulting from the crop according the encoder’s bounding box prediction? And what distance is being referred to - is it the translation offset t? I would have expected the bounding box itself to encode t. Overall this part of the paper was confusing to me. - How do you learn the encoder of 3D attributes such q and t? Do you assume there are ground truth labels? (Later in the implementation details it seems there are either ground truth labels or an approximately differentiable renderer is used. It may improve the clarity of the paper to explain this earlier, and give more details). - How important is it to use a GAN for texture synthesis? Since the network is trained as an auto-encoder, I do not see a need for multi-modality in the generator network, so in theory a simple deterministic model could be used, which might speed up training. - Fig 6: is there a reason that the background details change in the edited images? It would seem possible to copy pixels directly from the source for the regions that do not contain objects.

Reviewer 3



== Summary == This paper tackles the road scene manipulation problem by disentangling texture, semantic, and geometric information from a single image. A deep encoder-decoder network is introduced to learn such a disentangled representation. For each of the object instance, the method further infers the 3D object pose as its 3D-aware representation. Several image manipulation applications are demonstrated, thanks to a fully-disentangled 3D-aware representation. Experimental evaluations are conducted on KITTI and Cityscape benchmarks. == Quality and Originality == Overall, this paper is quite interesting and experimental results are promising. However, reviewer does have a few concerns about the proposed 3D-aware model. Hopefully, these concerns can be addressed in the rebuttal. == Object shape representation == Reviewer is a bit concerned as only one car model is used across all car instances (L169-L170). It does not sound like a principled approach for 3D disentangling in general. For example, the proposed approach is going to fail on object categories such as trucks, cyclists, or pedestrians. == Object occlusion == Reviewer is also concerned whether the proposed 3D-aware geometric inference method is robust to partial occlusions. For example, it might not be a good strategy to treat each car independently if one is occluded by the other. In some cases, the context information could help when there is partial occlusion. However, reviewer is not sure whether the current architecture (Figure 2: looks like the three streams are independent) addresses the issue mentioned here. == Indoor vs. Road Scene == As it is not shown in the main paper, reviewer would like to know the performance in indoor scene. This is a very critical point, as the title of the submission is about scene manipulation instead of road scene manipulation. == Missing references == References are incomplete regarding the interpretable 2D/3D representation in deep encoder-decoder networks. -- Attribute2Image: Conditional Image Generation from Visual Attributes, Yan et al. In ECCV 2016. -- Neural Face Editing with Intrinsic Image Disentangling, Shu et al. In CVPR 2017. -- Material Editing Using a Physically Based Rendering Network, Liu et al. In ICCV 2017. -- MoFA: Model-based Deep Convolutional Face Autoencoder for Unsupervised Monocular Reconstruction, Tewari et al. In ICCV 2017. -- Self-Supervised Intrinsic Image Decomposition, Janner et al. In NIPS 2017.